# Effectiveness of the Reasoning and Rehabilitation V2 Programme for Improving Personal and Social Skills in Spanish Adolescent Students

**DOI:** 10.3390/ijerph17093040

**Published:** 2020-04-27

**Authors:** Miriam Sánchez-SanSegundo, Rosario Ferrer-Cascales, Natalia Albaladejo-Blazquez, Raquel Alarcó-Rosales, Nicola Bowes, Nicolás Ruiz-Robledillo

**Affiliations:** 1Department of Health Psychology, Faculty of Health Science, University of Alicante, 03690 Alicante, Spain; miriam.sanchez@ua.es (M.S.-S.); rosario.ferrer@ua.es (R.F.-C.); alarco.rosales@gmail.com (R.A.-R.); nicolas.ruiz@ua.es (N.R.-R.); 2Cardiff School of Sport and Health Sciences, University of Cardiff, Cardiff CF23 6XD, UK; nbowes@cardiffmet.ac.uk

**Keywords:** reasoning and rehabilitation, intervention, skills, adolescents

## Abstract

Significant progress has been made in developing intervention programmes for adolescents at high risk of delinquency, school failure and emotional problems. The most effective programmes incorporate behavioural and skills training aimed at changing attitudes and promoting psychosocial and emotional skills in adolescents. This study examined the effectiveness of a school-based intervention programme based on the Reasoning and Rehabilitation V2 (R&R2). R&R2 is a cognitive behavioural programme developed using psychological theories about the aetiology of delinquency, as well as the cognitive, behavioural and socioemotional deficits in high-risk youth populations. A sample of 142 students (aged 13–17 years old) who were attending alternative education provision in Spain were randomly assigned to two experimental conditions (68 experimental group, 74 control group). The results showed that the R&R2 improved participants’ self-esteem, social skills, empathy and rational problem-solving with a medium–large effect size (η2 = 0.08 to 0.26). The effects of the programme were significant after controlling for age and the pre-test scores in baseline. These findings confirm the effectiveness of the Reasoning and Rehabilitation V2 programme in Spanish adolescent students and offer additional evidence regarding the implementation of the R&R2 programme in both alternative educational and mainstream school settings.

## 1. Introduction

Adolescence is a critical period of development characterised by the development of emotional competencies and cognitive abilities that are needed for independent functioning during adulthood [1]. The development of socioemotional components includes improved abilities to express and manage emotions as well as a gradual increase in response inhibition, abstract thinking and problem-solving.

In the last few decades, the effectiveness of a wide range of intervention programmes focused on training social and emotional skills in young people at high risk of school failure, delinquency, and drugs abuse have been developed [2,3,4]. Although there are debates around which intervention strategies are most effective, there is a consensus that early promotion of emotional skills and values are key factors for positive adult development [5,6]. Several studies have found that intervention programmes that incorporate behavioural and skills training aimed at changing attitudes and promoting social and emotional abilities during adolescence produce more beneficial outcomes than traditional intervention approaches [7,8,9].

The findings of cognitive behavioural interventions and adolescent development studies have driven the implementation of school-based interventions specifically designed to address deficits in emotional management and social skills, perhaps because these factors are linked to classroom climate and school achievement [8]. From this perspective, it has been noted that school-based interventions represent a promising approach to enhance success in school and increase interpersonal skills in adulthood [9]. A meta-analysis of school-based intervention programmes evaluating the impact of school programmes on social behaviour, problem behaviours and academic performance demonstrated that the largest effect sizes occurred for socioemotional skill performance (mean ES = 0.69). Compared with control groups, experimental groups demonstrated significantly improved social and emotional abilities, positive attitudes and academic performance that reflected an 11 percentile point gain in achievement. In addition, research has shown that those programmes based on cognitive behavioural models and those that commenced earlier in the school career were more effective in reducing delinquency and promoting positive behaviours in adulthood [10]. For example, a meta-analysis of 69 research studies carried out in America and Europe covering interventions for individuals at risk of delinquency and other risky behaviours (including 19 studies of interventions for adolescents) showed that behavioural and cognitive behavioural programmes were the most effective treatment programmes in both adult and adolescent samples, with a mean delinquency reduction for treated groups of about 30% [11].

To date, one of the most widely implemented cognitive behavioural programme is the “Reasoning and Rehabilitation” (R&R) programme [12,13], a cognitive skills programme that aims to address cognitive deficits and improve social and emotional skills in juvenile and adult population. The R&R programme includes a wide range of strategies to improve problem-solving, social perspective-taking, critical reasoning, empathy, negotiation skills and values [14] by using games, practical skills and facilitated discussions [15]. Since the original programme was designed in the 80s, many different versions have been developed and adapted in different countries, including North America, Canada, Australia, UK and Spain [5]. The R&R programme and the reduced version, R&R2, have demonstrated statistically significant improvements in social skills [6,16], self-esteem [17] and in peer relationships amongst young people who have committed offences [6]. A study examining the effectiveness of the R&R with an sample of imprisoned young people reported a medium to large effect size of improving participants’ self-esteem (η^2^ = 0.14) and social skills (η^2^ = 0.21), as well as a large effect size in reducing the rates of aggressiveness (η^2^ = 0.22) [18]. The R&R programme has also been found to be effective at reducing antisocial behaviour [18,19,20,21], reconvictions and rates of recidivism [6,21,22] in both juvenile and adult offenders. A meta-analysis conducted in four countries, involving sixteen evaluations of the utility of the R&R in clinical practice reported an overall effect size of r = 0.14 equivalent to a 14% reduction in re-offending for the treatment groups compared with controls [23,24]. Although the overall average of effect size was small in this meta-analysis, the studies examined used reconviction rates as main outcome reported; therefore, it is likely that these studies underestimated other possible benefits from the programme including cognitive thinking, violent attitudes, prosocial values and emotional skills [17]. Additionally, a meta-analysis of 57 European interventions showed that while the average effect of treatments on short-term in social skills was medium–large (r = 0.20), the average effect size for recidivism reduction was lower (r = 0.12) [25].

Despite the established evidence supporting use of the R&R programme for improving social and emotional skills and decreasing risk of delinquency, there is relatively limited research in non-offender populations. There is also a lack of studies on the utility of this programme in Spain, with only two studies, one of which evaluated the effectiveness of the R&R for reducing drug consumption [19], and the other for promoting social competence skills in a juvenile offenders population [16]. To our knowledge, no research in Spain has explored the impact the programme has on emotional and personal skills with non-offending adolescent students at high risk of failing school and attending alternative educational provision, labelled “basic professional training (FPB) and diversification programme (DP)”. These programmes constitute an alternative for those students who are interested in completing schooling but may not be well suited or able to engage or benefit within mainstream educational provision. Previous research has demonstrated that students who are not engaged in school, who are failing academically, or who feel disconnected from their schools and have emotional deficits are more likely to engage in risky behaviours including substance use and delinquency [26]. Therefore, the aim of this study was to assess the effectiveness of the R&R2 programme for improving personal and emotional skills in adolescent students attending FPB and DP (alternative education provision). It was hypothesised that the R&R programme would support students to improve emotional and social skills when compared with the control group.

## 2. Materials and Methods

### 2.1. Sample and Procedure

The initial sample comprised 183 students from Spain. Participants ranged in age from 13 to 17 years (M = 16.04, SD = 0.83). Inclusion criteria for the students were: (1) being part of an alternative educational programme (FPB or DP); (2) regular attendance in the classroom (at least 80% in the past 3 months); and (3) being able to read and complete the questionnaires on their own. Students were randomly assigned to two conditions using a cluster sampling design in two stages: schools were selected by probability to size sampling and random selection of classrooms with students 13 to 17 years old attending to alternative school programmes. From the initial sample, 41 participants were excluded. The remaining 142 students were assigned to the experimental group (n = 68; 54.5% females; 45.5% males) or control group (n = 74; 54.5% females; 45.5% males). Across the follow-up (Time 2 and Time 3), 50 students (32 experimental group and 41 control group) left the intervention programme and were classified as non-completers. These participants had either stopped attending school or refused to be part of the follow-up data collection. In total, 69 participants completed follow-up data collection across the two conditions (Figure 1).

The study was approved by the Ethics Committee of the University of Alicante (UA-2015-10-13). Participants were informed about the study and the process of withdrawing, and were asked to consent to participate. Consent was also taken from legal tutors and parents, in accordance with the Royal Decree 1720/2007 about Personal Data Protection. Participants were informed that their participation in the study was voluntary and that they could withdraw from the study with no consequences. Student or parental consents were obtained from 85% of the sample. For all students who did not return a signed form, several additional attempts were made by project staff. Parental refusal occurred in less than 5%.

### 2.2. Experimental Design

Participants were selected from eight similarly sized high schools offering alternative educational provision for students at high risk of failing school in Alicante (Spain). A total of 13 classes were identified (average class size 11 students) and classes were then assigned to one of two experimental conditions: experimental group (EG, n = 6) and waiting list control (CG, n = 7). Demographic data were gathered from participants including age, sex, nationality, academic attainment (having to repeat a course of education), economic resources, prior absenteeism at school and levels of anxiety, depression and stress. All participants completed a pre-/post-test and follow-up battery of questionnaires. For the experimental group, the pre-test battery was followed by the implementation of the R&R2 intervention programme over the subsequent 12 weeks. The programme was applied from January to March of the 2016 academic year. The programme consisted of twelve 2 h class sessions distributed across six months. A cut-off of ≥10 sessions of the 12 was used to classify students as completers (representing at least 80% attendance of the programme).

### 2.3. R&R2 Intervention Programme

The R&R2 [27] is a structured 12 to 15 session programme focused on training cognitive, attitudinal, emotional and behavioural characteristics that are associated with negative behaviours and mental health problems in youth and adult populations. Although the original R&R programme was targeted at medium- to high-risk offenders, the R&R2 is available for lower-risk offenders as well as for individuals who have not progressed toward illegal behaviour. The programme was developed for “*adolescents and young adults with impulse control problems who lack essential constructive planning, organizational and prosocial skills and values and are engaging in various disruptive and anti-social behaviors at home, in school, at work or in community*” (p.18) [27]. The programme offers a novel approach based on a body of evidence that the development of prosocial skills are associated with positive functioning. The R&R2 programme is implemented by qualified trainers who have received formal instruction from the programme on how to conduct the programme activities.

### 2.4. Measures

#### 2.4.1. RSES—Self-Esteem

The Rosenberg Self-Esteem Scale (RSES) is a 10 item self-report questionnaire assessing global self-esteem [28]. The RSES comprises 10 items (e.g., “I feel that I have a number of good qualities”). Participants rated the items on a 4 point Likert scale from 0 (strongly disagree) to 4 (strongly agree). For the present study, we used the Spanish adaptation, which has demonstrated adequate psychometric properties with alpha values of 0.83 [29]. In the present study, the items demonstrated good internal reliability (α = 0.85).

#### 2.4.2. SSS—Social Skills Scale

This scale is composed of 33 items (e.g., “I am unable to ask for discounts when I go to a store”) answered on a 4 point scale from 1 to 4, of which 28 are worded in an inverse sense and 5 are worded in a positive sense. The scale is comprised of six factors: “self-expression in social situations” (eight items); “defence of one’s rights as a consumer” (five items); “expression of anger or disagreement” (four items); “assertiveness” (six items); “making requests” (five items); and “starting interactions” (five items). It has been previously validated for the Spanish speaking population, showing adequate psychometric properties with alpha values ranging from 0.65 to 0.78 [30]. The scale showed good internal reliability with the current sample (α.69 to 0.81).

#### 2.4.3. SPSI-R—Social Problem-Solving Inventory, Revised

The SPSI-R (Social Problem-Solving Inventory, Revised) is a 52 item self-report measure that assesses people’s ability to solve problems in everyday living (e.g., “I believe that my problems can be solved”). Respondents rate each item on a 5 point Likert scale organised into five scales: “positive problem orientation” (PPO: 5 items), “negative problem orientation” (NPO: 10 items), “rational problem-solving skills” (RPS: 20 items), “avoidance style” (AS: 7 items) and “impulsivity/carelessness style” (ICS: 10 items). The SPSP-R has shown adequate psychometric properties with alpha values ranging from good (0.73) to excellent (0.95) for each scale [31]. In the present study, the items demonstrated good internal reliability with alpha values ranging from α = 0.69 to 0.85.

#### 2.4.4. IRI—Empathy

The Interpersonal Reactive Index [32] is a 28 item self-report questionnaire that assesses the construct of empathy under a multidimensional perspective (e.g., “Sometimes I don’t feel very sorry for other people when they are having problems). The scale is comprised of four factors covering the cognitive and emotional dimensions of empathy. The two cognitive scales include perspective taking, which assesses the tendency to adopt other’s point of views, and fantasy, which assesses the tendency to transport oneself imaginatively into fictitious characters and experience their emotions. The two emotional dimensions include empathic concern, which refers to feelings of sympathy and concern for others, and personal distress, which measures feelings of fear and discomfort. The items are answered on a 5 point scale, where 1 “does not describe me at all” and 5 “describes me very well”. The IRI has shown adequate psychometric properties with alpha values of approximately 0.70 for the Spanish-speaking population [33]. The scale had excellent internal reliability with our sample (α = 0.90).

### 2.5. Data Analysis

For the analyses, EG and CG scores of participants across all variables were gathered. Firstly, *t*-test comparison between EG and CG were employed to analyse possible differences in baseline between groups. Effect size estimation was also computed for each pair of variables using [34]. MANCOVA tests of repeated measures of ‘time’ (pre-and post-training assessments) with ‘group’ (participants from the EG vs. participants from the CG) as between-subject factor were performed to analyse the effectiveness of the intervention at improving personal and social skills. For significant results, partial eta-squared (η^2^) was reported as a measure of the effect size [35]. Age and pre-test scores age were introduced in the analyses as covariates. All statistical analyses were performed using SPSS (International Business Machines Corporation (IBM), Armonk, NY, USA), Statistics for Windows, Version 23.0, considering *p* < 0.05 to be significant. The descriptive values are expressed as mean and standard deviation (M and SD, respectively).

## 3. Results

### 3.1. Baseline Sociodemographic and Clinical Characteristics of Experimental and Control Groups

Baseline analysis of EG and CG in terms of the sociodemographic and clinical data is presented in Table 1. The experiment vs. control groups showed equivalent pre-treatment mean scores in sociodemographic domains except for age, where the control group were significantly younger than the experimental group. Both groups were equivalent in their reported levels of anxiety, depression and stress.

### 3.2. Effectiveness of the R&R2 Programme in Improving Self-Esteem

The EG (see Table 2) displayed a significant increase in mean score on the total scores of self-esteem, indicating a positive change in self-esteem attitudes in adolescent students. This effect was significant between Time 2 (post-intervention) and Time 3 (follow-up) (F = 16.6; *p* = 0.01; *η*^2^ = 0.19) even after controlling for age and the mean scores in Time 1 (pre-intervention).

### 3.3. Effectiveness of the R&R2 Programme for Improving Social Skills

A significant effect of the interaction group × time was found for all subscales of social skills across time (Table 2). When age and pre-test mean scores (T1) were introduced into the model as covariates, significant statistical effects were found between T2 and T3 for the total social skills score, showing a positive change in improved social skills in adolescent students (F = 29.12; *p* < 0.001; η^2^ = 0.30). This effect was also significant for all subscales: self-expression (F = 24.27; *p* < 0.001; η^2^ = 0.26); defence of rights (F = 16.36; *p* < 0.001; η^2^ = 0.19); disagreement (F = 13.46; *p* < 0.001; η^2^ = 0.17); assertiveness (F = 26.15; *p* < 0.001; η^2^ = 0.28); making requests (F = 13.08; *p* < 0.001; η^2^ = 0.16); and starting interactions (F = 2.733; *p* < 0.001; η^2^ = 0.16).

### 3.4. Effectiveness of the R&R2 Programme for Improving Empathy

Regarding empathy, significant differences between EG and CG were found across time, indicating a positive effect of the intervention programme on empathy in adolescent students. The interaction group × time was also significant between Time 2 (intervention) and Time 3 (follow-up) for all subdomains of empathy, even after controlling for age in statistical analysis: perspectives of others (F = 10.77; *p* < 0.001; η^2^ = 0.14); fantasy (F = 6.46; *p* < 0.001; η^2^ = 0.09); empathic concern (F = 6.39; *p* < 0.001; η^2^ = 0.09); and personal discomfort (F = 5.72; *p* < 0.001; η^2^ = 0.08).

### 3.5. Effectiveness of the R&R2 Programme for Improving Problem-Solving

The results of the impact of the R&R programme on problem-solving ability showed a significant effect of the interaction group × time for rational problem-solving, indicating a positive effect of the intervention between T2 and T3 in the EG vs CG groups. These results were significant after controlling for age and pre-test scores at baseline. However, no significant differences were found for the total scores for problem solving, positive and negative orientation, or impulsive and avoidant styles of problem solving.

## 4. Discussion

This study evaluated the effectiveness of the Reasoning and Rehabilitation Programme V2 on the promotion of emotional and social skills in adolescent students attending alternative school programmes in Spain. After controlling for age and baseline mean scores, students in the experimental condition reported an improvement in (medium–large effect size) self-esteem, social skills, empathy and rational problem-solving. The present results offer additional evidence regarding the implementation of the R&R programme in school settings and confirm their effectiveness outside of North America and Canada, where R&R research has been extensively conducted.

### 4.1. Effectiveness of the R&R Programme on Self-Esteem

Overall, the present findings show that the R&R2 programme improved self-esteem for the experimental group at both the post-intervention point (Time 2) and follow-up (Time 3) with a large effect size of 0.19. This finding is consistent with previous studies, which have reported that self-esteem and social interactions are necessary for prosocial adjustment and successful functioning [36]. Self-esteem in children and adolescents has been linked to improved social and interpersonal relationships, as well as higher levels of academic achievement [37]. By contrast, self-esteem deficits have been associated with a wide range of adjustment problems from childhood to adulthood including social isolation, substance abuse, academic problems, loneliness and limited opportunities to develop social skills. A meta-analytic review of 116 studies of changing self-esteem in children and adolescents demonstrated that the most effective programmes at changing self-esteem were those using a theoretical or empirical rationale, those using experimental vs. control groups and those focused on prevention and treatment of deficits in self-esteem. Our results were consistent with previous literature supporting the effectiveness of intervention programmes focused on promoting positive values and the promotion self-esteem.

### 4.2. Effectiveness of the R&R Programme on Social Skills

The R&R programme was also effective for improving social skills in the EG group from T2 to T3 with a large effect size, in line with our hypothesis. This effect was significant for all domains of the social skills scale. Studies examining the role of social skills on personal competences have demonstrated that social skills, defined as a set of specific behaviours that an individual exhibits to perform competently at a social task [38], are necessary for adjustment and successful interactions across the lifespan. The relationship between social skills and adjustment is bi-directional. For example, improved social skills competences can lead to better social interactions and an improved ability to communicate effectively with others. Social skills can also result in more frequent opportunities to engage social situations which can, in turn, increase the opportunities to develop positive interactions and maturation [39]. A meta-analysis of after-school programmes examining the effectiveness of personal and social skills in children and adolescents indicated that youth can benefit in multiple ways if these components are offered. Compared to controls, the experimental group demonstrated significant increases in positive social behaviours and academic achievement, and significant reductions in problem behaviours. In line with the implementation of the R&R programme, the presence of four practices were associated with effective training: sequenced, active, focused and explicit [10].

### 4.3. Effectiveness of the R&R Programme on Empathy

In line with our hypothesis, the empathy scale showed overall improvements for experimental group participants across time, with a medium to large effect size. The EG participants showed significant improvement in empathy, which involves processes that allow us to anticipate, understand and experience another’s point of view [40]. Past research has consistently demonstrated that empathy plays a key role in the development of positive relationships and altruistic behaviour [41]. Empathy has been considered as a set of cognitive and emotional components that predict the ability to view situations from another perspective, as well as to experience feelings of compassion, warmth and concern for others [36]. In a meta-analytic review of 213 school-based intervention programmes, the largest effect sizes for successful interventions were found for those programmes focused on training empathy, emotional recognition, problem solving and decision-making. In addition, those programmes focused on empathy that commenced earlier were more effective at promoting emotional and social skills in adolescent students [10].

### 4.4. Effectiveness of the R&R Programme on Problem Solving

Problem-solving has been defined as a general strategy by which a person attempts to discover or identify effective responses for a specific problem situation [30]. Problem-solving is important in social situations because it influences positive psychological adjustment [42]. The results of the current study supported our hypothesis in part. The current study suggested that the R&R2 programme was effective at improving the rational aspects of problem-solving. However, there were no significant effects of the programme in other factors, including positive, negative, impulsive and avoidant style scores at post-test among intervention and control groups. Research has demonstrated that the development of problem-solving is determined by the brain’s executive functioning, which plays a vital role in personal skills including strategic planning, abstract thinking, inhibitory control and problem-solving [43,44]. The development of executive functions begins around 12 months of age, but the maturation progress continues throughout the second decade of life, yielding a peak of development between the ages of 20 and 29 years [43]. The lack of effectiveness of the R&R programme on problem solving could perhaps be mediated by age of participants, given that students in our sample were considered cognitively immature to generate a variety of solutions, carefully anticipate the consequences of different problems and inhibit emotional responses when they fail to solve a problem in everyday life [10]. The gradual age-related increase in executive function ability from around 20 years of age has also been associated with an improved acquisition of inhibitory control, goal-directed behaviour and attentional set-shifting in the frontal brain regions, which are involved in maturation process across the life-spam.

### 4.5. Limitations and Clinical Research Implications

There were several limitations to the current study that may be helpful to consider in future research. The current study was conducted in a single city in Spain, and researchers and clinicians must therefore use caution when generalizing the findings to other areas of Spain or Europe. Secondly, outcome measures were gathered using self-reported data and it is possible that some adolescents may have underestimated or exaggerated their responses. Future studies should include additional evidence using teacher and family reports in order to triangulate responses between students, teachers and educators. Despite these limitations, this study provides solid evidence of the effectiveness of the Reasoning and Rehabilitation Programme V2 on the promotion of emotional and social skills in adolescent students. Future research should seek to replicate the study in Spain and also to explore the impact of R&R2 on adolescent populations studying a traditional academic curriculum in mainstream school settings.

The present results offer evidence regarding the implementation of the R&R programme in school settings, suggesting that, if these findings are confirmed in future studies, the R&R programme could be incorporated into routine educational practice. The current study has important implications for the development of empathy, social skills, problem-solving and self-esteem during adolescence. Past studies have suggested that implementing prosocial interventions focused on developing emotional and social skills in the beginning of adolescence may be one of the most effective ways to prevent early onset of socioemotional disturbances and risky behaviours throughout life [39,45]. Perception of personal skills has been considered an important variable influencing mental health adjustment [46]. Students who display high levels of personal skills show better social and interpersonal relations, psychological wellbeing and academic achievement [37]. As schools provide better educational opportunities for prosocial behaviours, the implementation of socioemotional interventions may influence the development of an unwritten curriculum defined as school culture, which promotes values, beliefs, norms, goals, sense of connection to the school and more positive expectations of students’ skills in school [47].

## 5. Conclusions

To our knowledge, this is the first study in Spain to explore the effectiveness of the Reasoning and Rehabilitation Programme V2 on the promotion of emotional and social skills in adolescent students. These findings extend the previous evidence of the effectiveness of the R&R programme and support the impact that school-based interventions may have in improving adolescent skills. As schools provide more opportunities for developing personal skills, R&R may contribute to increasing prosocial abilities by providing opportunities for adolescents to exercise emotional abilities such empathy and self-esteem, as well as social competencies including assertiveness and problem-solving. These abilities are an important part of adolescent development and relationships, and have an impact on school attainment, health and wellbeing.

## Figures and Tables

**Figure 1 ijerph-17-03040-f001:**
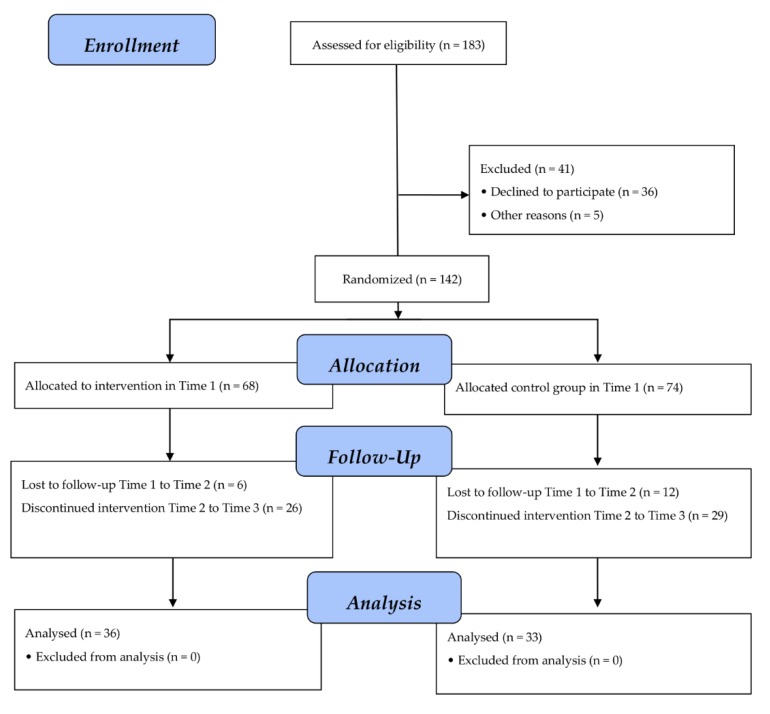
Consolidated Standards of Reporting Trials (CONSORT) flow diagram of participant recruitment.

**Table 1 ijerph-17-03040-t001:** Baseline sociodemographic and clinical characteristics of experimental and control groups.

	Control Group (n = 74)		Experimental Group (n = 68)			
	Men n (%)	Women n (%)	Men n (%)	Women n (%)	χ^2^	*p*
Sex	48 (64.9)	26 (35.1)	52 (76.5)	16 (23.5)	2.29	0.130
	Spanish n (%)	Other n (%)	Spanish n (%)	Other n (%)	χ^2^	*p*
Nationality	62 (83.8)	12 (16.2)	58 (85.3)	10 (14.7)	0.06	0.804
	Yes n (%)	No n (%)	Yes n (%)	No n (%)	χ^2^	*p*
Repeat a course	74 (100)	--	67 (98.5)	1 (1.5)	1.09	0.295
	Mean (SD)		Mean (SD)		t	*p*
Age	15.93 (0.60)		16.15 (0.63)		−2.07	0.040
Economic resources	17.97 (20.27)		12.29 (15.75)		1.83	0.066
Anxiety	3.39 (4.54)		3.69 (4.19)		−0.33	0.739
Depression	4.13 (5.16)		3.98 (4.37)		0.15	0.879
Stress	3.80 (4.09)		4.22 (4.69)		−0.45	0.654

**Table 2 ijerph-17-03040-t002:** Differences between EG and CG adolescents at post-intervention and 6 month follow-up.

Outcome	Group	Pre-Training		Post-Training		Follow-Up		Group Effect		
		Mean	SD	Mean	SD	Mean	SD	F	Direction	η^2^
**RSE—Self esteem**	Training	23.98	6.11	30.58	5.28	28.28	5.61	16.60 ***	Training > control	0.19
	Control	24.31	5.21	21.50	7.68	21.75	6.41			
**EHS—Social Skills**										
Self-expression	Training	15.53	5.39	25.18	4.26	21.45	4.84	24.27 ***	Training > control	0.26
	Control	16.53	5.51	15.56	5.90	16.72	6.33			
Defence of rights	Training	10.38	3.54	14.88	3.54	13.83	2.09	16.36 ***	Training > control	0.19
	Control	10.13	3.92	10.13	3.49	10.75	3.85			
Disagreement	Training	7.38	2.37	11.45	2.28	9.25	2.82	13.46 ***	Training > control	0.17
	Control	8.19	2.15	7.47	3.12	8.16	3.66			
Assertiveness	Training	11.45	3.76	18.40	2.46	14.68	3.79	26.15 ***	Training > control	0.28
	Control	12.06	4.48	11.03	4.66	11.75	4.71			
Making requests	Training	9.98	4.04	14.83	2.89	14.10	2.06	13.08 ***	Training > control	0.16
	Control	9.63	3.77	10.31	3.91	11.78	4.14			
Starting interactions	Training	10.35	3.93	15.03	2.46	14.70	2.80	13.16 ***	Training > control	0.16
	Control	8.97	3.27	9.88	3.86	11.53	4.46			
**IRI—Empathy**										
Perspectives of others	Training	15.05	4.69	19.33	4.29	15.53	4.18	10.77 ***	Training > control	0.14
	Control	15.22	4.01	14.25	3.96	12.81	3.47			
Fantasy	Training	12.05	4.52	16.50	4.77	14.53	5.00	6.46 **	Training > control	0.09
	Control	13.00	3.62	12.88	4.35	11.50	3.23			
Empathic concern	Training	13.53	4.47	17.10	4.03	16.08	5.52	6.39 **	Training > control	0.09
	Control	13.97	3.69	13.31	3.61	12.84	3.66			
Personal discomfort	Training	12.33	3.69	15.53	3.74	14.65	5.02	5.72 **	Training > control	0.08
	Control	13.41	4.07	12.69	3.44	11.84	3.07			
	Control	14.32	5.81	11.65	4.38	13.18	5.47			
**SPSI-R—Problem-Solving**										
Positive orientation	Training	9.48	5.58	11.53	4.41	10.78	6.18	3.83	No difference	0.05
	Control	8.59	5.17	6.34	5.46	7.78	5.41			
Negative orientation	Training	6.60	5.04	6.15	4.63	4.68	3.19	2.31	No difference	0.03
	Control	6.72	4.90	5.16	5.46	5.84	4.53			
Rational solution	Training	7.43	5.12	11.93	4.41	11.63	5.35	14.69 **	Training > control	0.18
	Control	7.56	5.17	6.31	5.79	7.00	4.50			
Impulsive style	Training	6.68	4.17	5.78	3.83	4.63	3.06	1.72	No difference	0.03
	Control	6.13	4.15	5.09	4.88	5.53	3.77			
Avoidance style	Training	5.65	3.73	5.90	3.76	5.60	3.39	0.16	No difference	0.00
	Control	6.97	4.74	5.44	5.19	5.19	4.19			

Statistical analyses adjusted for age and pre-test scores; ** *p* < 0.01; *** *p* < 0.001.

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
