# Peer review of "Effectiveness of the Reasoning and Rehabilitation V2 Programme for Improving Personal and Social Skills in Spanish Adolescent Students"

_ijerph, 2020, doi:10.3390/ijerph17093040_

Round 1

Reviewer 1 Report

Important topic: Trying to improve personal and social skills. How well does the program work.

Study can be added to a future meta-analysis.

Why did the authors choose to carry out a new empirical study instead of summarizing the existing literature in a meta-analysis? The sample size of the meta-analysis would be much larger than the sample size of the present study.

The text should definitely be checked by a native speaker of English; take out the massive number of typos. There are so many mistakes in the English that it detracts from the story the authors want to tell the readers. Journals clearly stipulate that papers should be written in correct English. Submitting papers in suboptimal English means the reviewers have to work harder and they can spend less time on improving the message of the manuscript, but have to wrestle through unclear sentences.

Why did the authors submit such a badly written manuscript?

I will give some examples of the many mistakes made, but they are illustrative, not exhaustive.

ABSTRACT

  1. 16 high students? They had been smoking marijuana? Please write correct English.

  1. 16 ‘attending to school programs’ should be ‘attending school programs’. A doctor attends to his patients. Write correct English.

There is a fundamental mistake in the Abstract: the effectiveness of the effect is described using significance testing, where one should report effect sizes. Please study carefully the new paper by Funder & Ozer (2019) [see below], who carefully describe the use of effect sizes. For instance, what are the effect sizes reported in meta-analyses on this topic or related topics, related training programs? Which effect is meaningful?

  1. 21 ‘the mean scores in baseline’? Write correct English.

The Abstract should begin by mentioning why the topic is relevant, and what kind of effects the authors are expecting. What is the research question? In Introduction, the authors mention that a large number of evaluation studies have been carried out, so they must be able to come up with a clear effect size to be expected.

  1. 24 ‘school implications’? Write correct English.

INTRODUCTION

Good review of the literature.

  1. 30 the maturation of components? Can components mature? Write correct English.
  2. 34 ‘high-risk’ should be ‘high risk’; why use a hyphen here? Write correct English.

l.49 should be ‘’ have shown

  1. 49 it is not interesting to learn that effects were significant; with a large enough sample size every effect becomes significant, even ridiculously small effects. For these kind of studies it is essential to know how much the behavior was changed, how much the skills were improved. Report the effect sizes.
  2. 52 Is it important to improve self-esteem? Isn’t this actually a bad thing? Self-esteem should be linked to achievement; better achievement should be followed by higher self-esteem; improving self-esteem absent improved of achievement seems a bad thing. Improving self-esteem by itself might lead to criminal behavior.
  3. 53 Is there maybe a meta-analysis summarizing the effects?
  4. 58 alternative high school programs? Unclear what is meant.
  5. 60 should read: interested in. Write correct English.
  6. 60 difficulties to get it? Get what? Write correct English.

METHOD

Solid Method section.

Sample sizes are acceptable, but clearly not large.

  1. 69 Again, what does it mean: Alternative school programs? Literally translated from Spanish?
  2. 71 ‘located at the East of Spain’ à located in the east of Spain. Write correct English.
  3. 71 participants ranged
  4. 81 ‘dropped out the school’? Write correct English.

RESULTS

Well done.

DISCUSSION

An important point for evaluations of effect sizes, how long are the effect sizes expected to last? Do the authors expect a fade-out effect?

An obvious question – a potential fundamental flaw in the paper – how did the authors deal with regression to the mean? This might potentially explain the whole effect. Problem students were selected, and these will be the most sensitive to regression to the mean.

  1. 210 ‘higher levels’; not really informative. Small effects, medium effects, strong effects?

Many topics are discussed: good.

What are the practical and theoretical implications? Especially the former is important for an applied study, such as the present one.

LITERATURE

Funder, D. C., & Ozer, D. J. (2019). Evaluating effect sizes in psychological research: Sense and nonsense. Advances in Methods and Practices in Psychological Science, 2, 156-168.

Author Response

REVIEWER 1

We are very thankful to you for giving us the opportunity of correcting our paper. We have responded point-by-point to the comments. Please find our responses below.

Why did the authors choose to carry out a new empirical study instead of summarizing the existing literature in a meta-analysis? The sample size of the meta-analysis would be much larger than the sample size of the present study.

Authors’ response: we thank the reviewer for this suggestion. We are interested in conducting intervention strategies to promote emotional and social skills for children and adolescents not only for research purpose but also for expanding our findings to the daily life in educational contexts. There is a lack of assessments in schools that spanned beyond the domains of academic functioning and mental health. Of course, it would be great in the future to summarize the existing literature in a meta-analysis.

The text should definitely be checked by a native speaker of English; take out the massive number of typos. There are so many mistakes in the English that it detracts from the story the authors want to tell the readers. Journals clearly stipulate that papers should be written in correct English. Submitting papers in suboptimal English means the reviewers have to work harder and they can spend less time on improving the message of the manuscript, but have to wrestle through unclear sentences. Why did the authors submit such a badly written manuscript? I will give some examples of the many mistakes made, but they are illustrative, not exhaustive.

 Authors’ response: thank you very much for checking the grammar. Paper was previously checked by a native speaker of English which is an author of the paper. However, it has been checked thoroughly again. We hope all errors have been detected and changed.

ABSTRACT

Line 16 high students? They had been smoking marijuana? Please write correct English.

Authors’ response: It has been checked thoroughly again. We hope all errors have been detected and changed.

There is a fundamental mistake in the Abstract: the effectiveness of the effect is described using significance testing, where one should report effect sizes. Please study carefully the new paper by Funder & Ozer (2019) [see below], who carefully describe the use of effect sizes.

Authors’ response: Thanks for the reference of the above paper. Now, results in terms of effect size have been reported in the abstract and across the different sections in terms of eta square. Thank you for your suggestion

Line 21 ‘the mean scores in baseline’? Write correct English.

Authors’ response: It has been deleted.

The Abstract should begin by mentioning why the topic is relevant, and what kind of effects the authors are expecting. What is the research question? In Introduction, the authors mention that a large number of evaluation studies have been carried out, so they must be able to come up with a clear effect size to be expected.

Authors’ response: the abstract has been modified in the revised version.

Line 24 ‘school implications’? Write correct English.

Authors’ response: Thank you. It has been changed.

INTRODUCTION

Good review of the literature.

 Authors’ response: Thank you very much for your positive feedback.

Line 30 the maturation of components? Can components mature? Write correct English.

Line 34 ‘high-risk’ should be ‘high risk’; why use a hyphen here? Write correct English.

Line 49 should be ‘’ have shown

Authors’ response: Thank you. It has been changed.

Line 49 it is not interesting to learn that effects were significant; with a large enough sample size every effect becomes significant, even ridiculously small effects. For these kind of studies it is essential to know how much the behavior was changed, how much the skills were improved. Report the effect sizes.

Authors’ response: Thank you. It has been added as indicated in lines 53 and 57 to 76, 87.

Line 52 Is it important to improve self-esteem? Isn’t this actually a bad thing? Self-esteem should be linked to achievement; better achievement should be followed by higher self-esteem; improving self-esteem absent improved of achievement seems a bad thing. Improving self-esteem by itself might lead to criminal behavior.

Authors’ response:  The R & R2 program is a cognitive-behavioral and multimodal program that focuses on training social skills, moral values, forms of critical thinking and prosocial thinking. All trained skills within the program are interrelated. Thus, the way in which participants perceive themselves, the way they think and solve their problems, plays an important role in behavior, and especially in personal, emotional and social adjustment. The positive effect of the R & R2 program for the improvement of positive self-esteem is relevant, due to that in absence of a positive and realistic assessment of oneself it is very difficult to overcome obstacles and social conflicts. Also, according to different studies, an adequate level of self-esteem is necessary for showing respect and experience favorable feelings towards others (Martínez-Otero 1999, 2000, 2003). A recent longitudinal study examining the relations in self-esteem and prosocial behavior toward other showed that adolescent self-esteem was associated longitudinally with subsequent prosocial behavior toward strangers, and earlier prosocial behavior toward strangers promoted subsequent self-esteem (Fu, Padilla-Walker, & Brown, 2017).

REFERENCES

Fu, X., Padilla-Walker, L. M., & Brown, M. N. (2017). Longitudinal relations between adolescents' self-esteem and prosocial behavior toward strangers, friends and family. Journal of adolescence, 57, 90-98.

Line 53 Is there maybe a meta-analysis summarizing the effects?

Authors’ response: This paragraph has been modified.

Line 58 alternative high school programs? Unclear what is meant.

Authors’ response: A native English speaking author has revised this sentence and it has been modified as “alternative education provision”

Line 60 should read: interested in. Write correct English.

Line 60 difficulties to get it? Get what? Write correct English.

 Authors’ response: Thank you. It has been changed (see in yellow).

METHOD

Solid Method section.

 Authors’ response: Thank you very much for your positive feedback.

Sample sizes are acceptable, but clearly not large.

Line 69 Again, what does it mean: Alternative school programs? Literally translated from Spanish?

Line 71 ‘located at the East of Spain’ à located in the east of Spain. Write correct English.

71 participants ranged

 Authors’ response: Thank you very much for your help and your revision. It has been changed (see in yellow).

Ok, thank you so much

Line 81 ‘dropped out the school’? Write correct English.

  Authors’ response: Thank you very much for your help and your revision. It has been changed (see in yellow).

RESULTS

Well done.

 Authors’ response: Thank you very much for your positive feedback.

DISCUSSION

An obvious question – a potential fundamental flaw in the paper – how did the authors deal with regression to the mean? This might potentially explain the whole effect. Problem students were selected, and these will be the most sensitive to regression to the mean.

 Authors’ response: Prior research has suggested possible strategies for controlling regression to the mean including the control of sources of intra-individual variability such as physiological and psychological variability, changes in measurement conditions, and random measurement errors (Guallar, Jiménez, Garcia-Alonso, & Bakke, 1998). Participants in our study were matched at baseline in intra-individual psychological variables (levels of depression, anxiety and stress) and significant variables at baseline such as age and pre-test mean scores which were introduced in the model as a covariates to control sources of variability between groups. Additional strategies for controlling regression to the media have been proposed and were controlled in this study including the standardization of measurement, the use of trained personnel, the certification and validation of measurement instruments, and the use of accurate tests. These strategies have been proposed as good clinical practices that contribute to reducing the variability of the determinations and, therefore, the effect from regression to mean (Guallar et al., 1998). Also it has been suggested that in clinical trials, the inclusion of a control group directly avoids bias in the estimation of the effect produced by the regression to the mean, since both the control and intervention group suffer the same degree of regression and, therefore, any difference between both groups will be attributable to the effect of the intervention (Davis, 1976). The existence of a control group, however, does not avoid the classification errors and the dilution that these produce in the effect estimates. An effective strategy is to improve the selection process by taking the mean of a series of initial measurements as the baseline determination for each patient or group. We controlled the differences between groups in the pre-test to control sources of variability between groups in the baseline.

REFERENCES

Guallar, E., Jimenez, F. J., García-Alonson, F., & Bakke, O.M. (1997). La regression a la media en la investigación y práctica clínica. Medicina clínica, 109, 23-26.

Davis, C.E. (1976). The effect of regression to the mean in epidemiologic and clinical studies. American journal of epidemiology, 104(5), 493-498.

Line 210 ‘higher levels’; not really informative. Small effects, medium effects, strong effects?

Authors’ response: Thank you. It has been changed (see in yellow).

DISCUSSION

Many topics are discussed: good.

 Authors’ response: Thank you very much for your positive feedback.

What are the practical and theoretical implications? Especially the former is important for an applied study, such as the present one.

 Authors’ response: Now, we have included implications in lines 338 to 351.

Reviewer 2 Report

Thank you for sending me the manuscript to review.

The main limitation is found in the Introduction, the theoretical revision is scarce, superficial and little updated. They should expand this section by including more relevant and current research papers on the subject.

Where are the research hypotheses?

Another section that should be reviewed by the authors is the method. In the first subsection, they are mixing content on ethical aspects of the research and description of the sample. For example lines 68-71 should be written together with the reference to informed consent.
Regarding the description of instruments, they should include an example of an item for each scale or subscale. In addition, they must provide data on the reliability obtained with their study sample.

Finally, the discussion shows the limitations of the Introduction section, therefore it should be reformulated, once the theoretical revision part has been improved. Also taking into account the contrast of hypotheses that will have been previously formulated.

Author Response

REVIEWER 2

We are very thankful to you for giving us the opportunity of correcting our paper. We have responded point-by-point to the comments. Please find our responses below.

The main limitation is found in the Introduction, the theoretical revision is scarce, superficial and little updated. They should expand this section by including more relevant and current research papers on the subject.

Where are the research hypotheses?

We are very thankful to you for giving us the opportunity of correcting our paper. We have responded point-by-point to the comments. Please find our responses below.

 Authors’ response: There are conflicting opinions among reviewers with respect to the introduction section. While reviewer 1 suggest “Good review of the literature in the introduction section”, reviewer 2 suggest that theoretical revision is scarce and superficial. We have modified the introduction section according to your suggestion by including additional review of the literature in this section. Changes can be seen in yellow. Also, we have included research hypotheses at the end of introduction.

Another section that should be reviewed by the authors is the method. In the first subsection, they are mixing content on ethical aspects of the research and description of the sample. For example lines 68-71 should be written together with the reference to informed consent.

 Authors’ response: We agree. It has been modified in the revised manuscript

Regarding the description of instruments, they should include an example of an item for each scale or subscale. In addition, they must provide data on the reliability obtained with their study sample.

 Authors’ response: It has been added in the revised manuscript.

Finally, the discussion shows the limitations of the Introduction section, therefore it should be reformulated, once the theoretical revisions part has been improved. Also, taking into account the contrast of hypotheses that will have been previously formulated.

Authors’ response: There are conflicting opinions among reviewers with respect to the discussion section. While reviewer 1 suggest “Good discussion where many topics have been discussed”, reviewer 2 suggest that this section should be reformulated. We disagree with reviewer 2 given that we provide a rational discussion point-by-point to the findings found in our paper and we provide additional evidence that support our findings. We have of course, discuss the implications in a new paragraph and tested the added hypotheses. Hoping the reviewer 2 is agree.

Round 2

Reviewer 1 Report

The authors did a good job processing my detailed feedback. My compliments.

I am surprised that the first version of the manuscript was actually checked by a native speaker of English, as there were so many mistakes in it.

Author Response

Thank you so much for all your help and your positive feedback. It has been checked again. Thank you

Reviewer 2 Report

Please check some errata in the text, for example in 3.3. it says "emphaty" instead of "empathy". It would be nice if the authors could review the entire manuscript to detect possible errors and correct them.

Thanks

Author Response

Thank you very much for all your help to improve the current version of the manuscript. It has been checked again by a native english speaking who is author of the manuscrip.